# A Meta-Analysis of the Influence of Antipsychotics on Cytokines Levels in First Episode Psychosis

**DOI:** 10.3390/jcm10112488

**Published:** 2021-06-04

**Authors:** Piotr Marcinowicz, Magdalena Więdłocha, Natalia Zborowska, Weronika Dębowska, Piotr Podwalski, Błażej Misiak, Ernest Tyburski, Agata Szulc

**Affiliations:** 1Department of Psychiatry, Faculty of Health Sciences, Medical University of Warsaw, 02-091 Warsaw, Poland; piotr.marcinowicz@gmail.com (P.M.); mgdwiedlocha@gmail.com (M.W.); w.debowska@gmail.com (W.D.); agataszulc@poczta.onet.pl (A.S.); 2Department of Psychiatry, Pomeranian Medical University in Szczecin, 70-204 Szczecin, Poland; piotr.podwalski@gmail.com; 3Department of Psychiatry, Wroclaw Medical University, 50-367 Wrocław, Poland; mblazej@interia.eu; 4Institute of Psychology, SWPS University of Social Sciences and Humanities, 03-815 Poznań, Poland; etyburski@swps.edu.pl

**Keywords:** schizophrenia, cytokines, first episode psychosis, immune activation, psychiatry

## Abstract

Background: Cytokines have a major impact on the neurotransmitter networks that are involved in schizophrenia pathophysiology. First Episode Psychosis (FEP) patients exhibit abnormalities in cytokines levels prior to the start of treatment. Previous studies showed that antipsychotic treatment modulates cytokines levels. The aim of this meta-analysis is to further investigate this relationship. Methods: Several online databases were searched. For meta-analysis of selected studies, we analysed variables containing the number of cases, mean and standard deviation of IL-1β, IL-2, IL-4, IL-6, IL-10, IL-17, TNF-α, IFN-γ levels before, and after, antipsychotic treatment. Results: 12 studies were included in the meta-analysis. Our main results demonstrate that, in FEP patients, antipsychotic treatment is related to decreased concentrations of pro-inflammatory IL-1β, IL-6, IFN-γ, TNF-α and anti-inflammatory IL-4, IL-10 cytokines. On the other hand, levels of pro-inflammatory IL-2 and IL-17 remain unaffected. Conclusions: When compared with other meta-analyses of studies involving FEP individuals, results we obtained are consistent regarding decrease in IL-1β, IL-6. Comparing outcomes of our study with meta-analyses of schizophrenic subjects, in general, our results are consistent in IL-1β, IL-6, TNF-α, IFN-γ, IL-2. Our meta-analysis is the only one which indicates a decrease in anti-inflammatory IL-10 in FEP patients after antipsychotic treatment.

## 1. Introduction

Immune system dysfunction is a well-established factor in schizophrenia pathogenesis. Influenza, infection with Toxoplasma gondii, and Herpes Simplex Virus Type-2 during pregnancy are known risk factors that contribute to later development of schizophrenia in the offspring [1,2,3]. Animal and human studies confirmed that lymphocyte T cells transmigrating to the central nervous system (CNS) are responsible for cytokines release, neurogenesis modulation, and cognitive deficits, as well as altered behavior [4,5]. The lymphocyte T helper cells (Th cells), also known as CD4+ cells, are a type of T cells that have an important function in the immune system, mainly through the release of cytokines. Cytokines and their receptors are a group of regulatory proteins with critical impact on processes related to immunomodulation and inflammation, which influences the central nervous system (CNS) [6]. Proinflammatory cytokines-interleukin 2 (IL-2), IL-12, interferon (IFN)-γ and tumor necrosis factor (TNF)-α take part in the immune response type 1—cellular immunity for which Th1 cells are responsible [6]. On the other hand, proinflammatory IL-6 and anti-inflammatory IL-4, IL-10, and IL-13 take part in humoral immunity (immune response type 2) modulated by Th2 cells [6]. Additionally, a third subtype of CD4+ T cells is distinguished-Th17. It produces a proinflammatory IL-17 cytokine [6]. Studies showed that it plays an important role in neuroinflammation and exhibits neurotoxic properties [7]. Additionally, another process in which IL-1β, IL-6, and TNF-α are released in CNS is microglia activation. It contributes to apoptosis, inhibition of neurogenesis, and white matter abnormalities in the brain of patients with onset and relapse stages of schizophrenia [8,9,10].

It was proved that cytokines have a major impact on neurotransmitter networks that are involved in schizophrenia pathophysiology. IL-6 decreases the survival rate of serotoninergic neurons in the brain of rat’s fetus [11]. Potter et al. showed that IL-1β induces rat mesencephalic progenitor cells to convert into a dopaminergic phenotype [12]. A relationship between chronic administration of IFN-α and decreased release of dopamine, resulting in anhedonia, was observed in animals’ striatum [13]. It was also shown that cytokines influence tryptophan metabolism in the kynurenine pathway [14]. Proinflammatory cytokines, such as IL-6, IFN-γ, TNF-α, IL-1β, IL-2, and IL-18 are known inductors of indoleamine 2,3-dioxygenase (IDO)-1 expression. This enzyme takes part in tryptophan transformation into kynurenine, which is converted in astrocytes into a N-methyl-D-aspartate (NMDA) receptor antagonist-kynurenic acid (KYNA). NMDA hypofunction is also involved schizophrenia pathophysiology [14]. Elevated KYNA levels and increased IDO activity were observed in schizophrenia subjects.

Abnormalities in cytokine levels are present in patients diagnosed with schizophrenia, their first-degree relatives and first episode psychosis (FEP) individuals [15,16]. There are several operational definitions of FEP, which might be misleading and cause inconsistencies in conducted studies. It may be discussed in three categories: as first treatment contact, by duration of psychosis, or duration of antipsychotic treatment [17]. For the scope of this review we assumed that FEP is a distinct episode of psychotic symptoms with early onset that has occurred for the first time in an antipsychotic-naïve subject.

A meta-analysis carried out by Miller et al. showed that in comparison with healthy controls (HC) FEP subjects exhibited elevated levels of the following markers: IL-1β, IL-6, IL-12, IFN-γ, TNF-α, transforming growth factor (TGF)-β and the soluble interleukin-2 receptor (sIL-2R). No differences in IL-2 concentrations were observed [16]. Increased levels of IL-6, IFN-γ, TNF-α, and TGF-β were also noted in acute relapsed schizophrenia inpatients (AR). AR patients exhibited higher concentrations of IL-8 and IL-1RA and lower of IL-10 [16]. Results of a meta-analysis performed by Goldsmith et al. largely confirmed the findings of Miller et al. [18]. Both studies showed that FEP patients have higher levels of IL-1β, IL-6, IL-12, IFN-γ, TNF-α, TGF-β, sIL-2R, and no difference in concentration of IL-2 in comparison with HC [16,18]. Additionally, the Goldsmith et al. study showed that FEP subjects exhibit elevated levels of the interleukin-1 receptor antagonist (IL-1RA) and IL-10, lowered IL-4 and no difference in IL-17 and IL-18 in comparison with control groups. Differences in cytokine levels were similar in AR patients’ group, whereas only IL-10 levels were lower when compared to the control group, which is similar to Miller et al. [16,18]. A recent meta-analysis performed on 59 studies, comprising of 3002 FEP subjects confirmed increased levels of IL-6 and TNF-α when compared to HC. What is more, the authors failed to establish any relationship between higher levels of these proinflammatory cytokines and mean age, sex, diagnosis, BMI, tobacco/cannabis/other substance use or abuse, exclusion of medical comorbidity, or exclusion of anti-inflammatory treatment [19]. These findings further imply a relation between IL-6, TNFα and psychosis occurrence.

As studies discussed above show, FEP patients exhibit abnormalities in cytokines levels prior to the start of treatment. Feigenson et al. suggested that it may constitute a biochemical endophenotype in certain schizophrenia populations [20]. The importance of immune dysfunctions as a clinical marker was confirmed by the OPTiMiSE study—a largescale trial of antipsychotic response in FEP subjects [21]. Researchers identified a subtype of FEP individuals who exhibited the most severe symptoms and were at the highest risk of being non-responders when treated with amisulpride. Prior to antipsychotic treatment this subgroup had elevated serum levels of several pro-inflammatory cytokines and inflammation-associated biomarkers when compared to other patients [21].

Meta-analyses of studies, evaluating the influence of antipsychotic drugs on cytokines and their receptors, show antipsychotic treatment-induced changes in levels of inflammation markers. However, the scope of affected biomarkers, and their influence on pathogenesis and the clinical state of schizophrenia, are still a subject of research [16,18,22,23]. Evaluating the impact of antipsychotic treatment on inflammation markers in FEP individuals is of particularly high value, as it enables us to exclude the influence of a chronic psychotic process, as well as prolonged antipsychotic treatment and its side effects. Capuzzi et al. and Romeo et al. presented meta-analyses of studies measuring levels of cytokines prior to, and after, antipsychotic treatment in the specific population of FEP subjects. These analyses were conducted on a relatively low number of studies—7 [22] and 8 [24]. Their results were not fully convergent. Both Capuzzi et al. and Romeo et al. showed a significant impact of antipsychotic treatment on the decrease in pro-inflammatory cytokines IL-1β and IL-6. There was no effect on concentrations of TNF-α, IFN-γ. Results of these meta-analyses vary with regard to impact on IL-2, IL-4, and IL-17 [22,24]. The aim of this review and meta-analysis is to synthetize up-to-date findings and further investigate the influence of antipsychotics on the cytokine levels in FEP individuals. This meta-analysis was conducted in accordance with the PRISMA 2020 guidelines [25].

## 2. Materials and Methods

### 2.1. Systematic Search and Criteria

Two reviewers (W.D. and N.Z.) independently searched available online databases (PUBMED, PsychInfo, Scopus, Medline) for relevant studies published until 1 December 2020. The primary search terms were similar to those used by Capuzzi et al.:

PubMed: (“Schizophrenia”[Mesh] OR psychosis) and (“Antipsychotic OR antipsychotic* [title/abstract]) and (“Cytokines”[Mesh] OR cytokine*[title/abstract]);

Scopus: TITLE-ABS-KEY (antipsychotic* AND cytokines AND (schizophrenia OR psychosis));

PsycInfo (via Proquest): (cytokines) AND (schizophrenia OR antipsychotic* OR psychosis).

The search was double-checked by P.M. and broadened through addition of relevant synonyms and associated phrases such as: interleukine*, proinflammatory, inflammation, neuroleptic, TNF, IFN. Additional studies were identified by cross-referencing.

In contrast to Capuzzi et al. [24] we included studies measuring cytokines, not only at 4 weeks, but at different time-points to provide a broader dataset. A study was considered as eligible when: (1) it was published in English in a peer reviewed journal, (2) included FEP adults with no significant comorbidities, treated with neuroleptic medication, and (3) cytokine measurements were provided at two time-points as means + standard deviation (SD). Studies that did not meet all the inclusion criteria were excluded. The data (general study information and characteristics, means + standard deviation (SD) of cytokines levels) from selected studies was extracted by N.Z. and double-checked by P.M. and A.S. independently. All selected studies were assessed for risk of bias by P.M. and N.Z. independently, then consulted by A.S. (see Figure 1). For risk of bias evaluation, the Cochrane method ROBINS-I tool was used [26].

### 2.2. Statistical Methods

Data analyses were performed using RevMan, version 5.3 [27]. For meta-analysis of selected studies with a continuous measure, we analyzed variables provided in at least 3 studies, containing the number of cases, mean, and standard deviation of the level of IL-1β, IL-2, IL-4, IL-6, IL-8, IL-10, IL-12, IFN-γ, TNF-α observed before and after antipsychotic treatment. Cohran’s Q statistics for estimation of heterogeneity of studies was used. *p* value of 0.05 was used as a cut-off for significance [28]. When heterogeneity was present, the random effect model was employed to calculate standardized mean difference. I^2^ values of 25, 50, and 75 were considered indicative of mild, moderate, and marked heterogeneity [29]. The data from the Borovcanin et al. [30] study for responders and non-responders were pooled together in accordance with Cochrane 2011 methods manual [31]. As for results of Juncal-Ruiz et al. [32] the study arms were considered separately due to different drugs used in each arm.

Some studies, or measurements of specific cytokines, reporting relevant data were excluded from our meta-analyses. Excluded studies met at least one of three possible criteria:

Low data quality. This criterion mainly includes missing data and results not published in English. It was assumed that they would not be considered or substituted by statistically computed values on the basis of other studies.Several sensitivity analyses were performed after excluding each individual study (only one study each time). Studies shown to influence the overall result were excluded. This method was applied in the case of 3 or more analyzed studies.The study was excluded if its rejection abolished the publication bias. We created funnel plots to check for the existence of publication bias. This method was applied in case of 5 or more analyzed studies.

## 3. Results

### 3.1. Characteristics of Included Studies

12 studies were included in our meta-analysis (see Table 1 and Table 2). All studies were published in English language and in peer reviewed journals. The study duration ranged between 4 and 24 weeks on a sample of 24 to 83 FEP subjects. A variety of cytokines was measured by the researchers enabling a meta-analysis of the neuroleptics’ effect on IL-1β, IL-2, IL-4, IL-6, IL-10, IL-17, IFN-γ, TNF-α. Five studies evaluated treatment with risperidone alone. Two of the studies utilized patient plasma for evaluating cytokines, the rest used blood serum. Exclusion of these 2 studies in sensitivity analysis did not influence the overall results.

#### 3.1.1. IL-10

Results from 3 studies (*n* = 168/150) were included in the meta-analysis of IL-10 (Figure 2). Using the fixed and random-effects model, we did find a significant decrease in IL-10 levels after treatment (*p* = 0.02). Heterogeneity among studies was marked (I^2^ = 68%). Available data was not sufficient to perform comparisons with control groups. Exclusion of Juncal-Ruiz et al. [32], as well as the Noto et al. [40] study, rendered the results insignificant.

#### 3.1.2. IL-17

Results from 3 studies (*n* = 203) were included in the meta-analysis of IL-17 (Figure 3). Using the fixed and random-effects model, we did not find any conclusive changes in IL-17 levels (*p* = 0.51). Heterogeneity among studies was high (I^2^ = 93%). Neither baseline nor after treatment levels of IL-17 were significantly different from those of HC. Sensitivity analysis did not show any change of result after exclusion of each individual study.

#### 3.1.3. IL-1β

Results from 7 studies (*n* = 294/276) were included in the meta-analysis of IL-1β (Figure 4). Using the fixed and random-effects model, we did find a significant decrease in IL-1β levels after treatment (*p* = 0.006). Heterogeneity among studies was marked (I^2^ = 74%). After exclusion of the two main heterogeneity contributing studies (Azizi et al. [38]; Subbanna et al. [35]) the results provided the same conclusion. At baseline measurements, levels of IL-1β were significantly higher (*p* = 0.02) than HC, but after treatment, this elevation became statistically insignificant (*p* = 0.13). Sensitivity analysis did not show any change of result after exclusion of each individual study.

#### 3.1.4. IL-2

Results from 4 studies (*n* = 163/145) were included in the meta-analysis of IL-2 (Figure 5). Using the fixed and random-effects model, we did not find significant changes of IL-2 levels after treatment (*p* = 0.87). Heterogeneity among studies was high (I^2^ = 96%). Available data was not sufficient to perform comparisons with control groups. Sensitivity analysis did not show any change of result after exclusion of each individual study.

#### 3.1.5. TNF-α

Results from 7 studies (*n* = 346/328) were included in the meta-analysis of TNF-α (Figure 6). Using the fixed and random-effects model, we did find a significant decrease in TNF-α levels after treatment (*p* = 0.02). Heterogeneity among studies was high (I^2^ = 76%). At baseline measurements, the levels of TNF-α were significantly higher (*p* = 0.0002) than HC, but after treatment, this elevation became statistically insignificant (*p* = 0.09). Individual exclusion of Azizi et al. [38], Juncal-Ruiz et al. [32], Noto et al. [40] studies rendered the results insignificant.

#### 3.1.6. IFN-γ

Results from 5 studies (*n* = 261/243) were included in the meta-analysis of IFN-γ (Figure 7). The Mondelli et al. study was excluded from meta-analysis due to sample heterogeneity [42]. Using the fixed and random-effects model, we did find a significant decrease in IFN-γ levels after treatment (*p* < 0.00001). Heterogeneity among studies was moderate (I^2^ = 49%). At baseline measurements, levels of IFN-γ were significantly higher (*p* < 0.00001) than HC and the elevation after treatment remained statistically significant (*p* < 0.0001). Sensitivity analysis did not show any change of result after exclusion of each individual study.

#### 3.1.7. IL-6

Results from 8 studies (*n* = 427/409) were included in the meta-analysis of IL-6 after exclusion of 2 studies (Mondelli et al. and Petrikis et al.) due to sample heterogeneity (Figure 8). Using the fixed and random-effects model, we did find a significant decrease in IL-6 levels after treatment (*p* = 0.0001). Heterogeneity among studies was marked (I^2^ = 73%). If the two excluded studies were to be included, the significance of the results would be lost. At baseline measurements, the levels of IL-6 were significantly higher (*p* < 0.00001) than HC and the elevation after treatment remained statistically significant (*p* = 0.0004). Sensitivity analysis did not show any change of result after exclusion of each individual study.

#### 3.1.8. IL-4

Results from 4 studies (*n* = 168/150) were included in our meta-analysis of IL-4 after exclusion of Borovcanin et al. due to sample heterogeneity (Figure 9). Using the fixed and random-effects model, we did find a significant decrease in IL-4 levels after treatment (*p* = 0.01). Heterogeneity among studies was moderate (I^2^ = 41%). At baseline measurements, the levels of IL-4 were significantly higher (*p* = 0.02) than HC, but after treatment this elevation became statistically insignificant (*p* = 0.17). Individual exclusion of Noto et al. [40] and Haring et al. [33] studies rendered the results insignificant.

### 3.2. Risk of Bias

Overall risk of bias was assessed as moderate. The analysis of funnel plots for IL-1β, IL-6, IFN-γ, and TNF-α suggests some risk of publication bias. The results of ROBINS-I bias evaluation are presented in Table 3.

## 4. Discussion

Our results demonstrate that, in FEP patients, antipsychotic treatment is related to decreased concentration of pro-inflammatory IL-1β, IL-6, IFN-γ, and TNF-α and anti-inflammatory IL-4, IL-10 cytokines. On the other hand, levels of pro-inflammatory IL-2 and IL-17 remain unaffected. The exclusion of studies including subjects with history of prior treatment with antipsychotic medications had no significant impact on IL-1β, IL-6, and IFN-γ results, while rendering decreased TNF-α levels statistically insignificant. We also showed that baseline levels of IL-1β, IL-4 and TNF-α in FEP subjects were considerably higher in comparison to HC. After treatment, the levels of these cytokines were similar to those in HC. However, IL-6 and IFN-γ concentrations in FEP subjects were notably higher than in HC both before and after antipsychotics administration. Despite a decrease in the concentration of these pro-inflammatory cytokines, their levels remained elevated. The results that we obtained confirm earlier reports indicating elevated concentrations of IL-1β, IL-6, IFN-γ, and TNF-α in FEP subjects [16,18,19]. Significantly elevated levels of macrophage-derived cytokines IL-1β, IL-6, and TNF-α, as well as the Th1-derived cytokine IFN-γ support the macrophage-T-lymphocyte theory of schizophrenia pathophysiology.

Meta-analyses, evaluating the influence of antipsychotic treatment, on levels of cytokines, and their receptors, in subjects diagnosed with schizophrenia confirm anti-inflammatory effects of neuroleptic drugs to a varying degree. In Miller et al., meta-analysis the impact of neuroleptics was assessed based on 40 studies involving FEP and AR patients, and it showed an increase in sIL-2R and IL-12 and a decrease in IL-1β, IL-6, and TGF-β. Antipsychotic treatment had no effect on IFN-γ and TNF-α. These cytokines, along with sIL-2R, were described as “trait markers” which stems from the fact that they were elevated both before and after treatment [16]. On the other hand, IL-1β, IL-6, and TGF-β were identified as state markers, because their concentration significantly lowered after antipsychotic treatment [16]. The Toujrman et al. meta-analysis indicated increased levels of IL-12 and sIL-2R and a decrease in IL-1β and IFN-γ. No change in concentration of other measured markers (IL-2, IL-4, IL-6, IL-10, IL-1RA, sIL-6R, TGF-β, TNF-α) was observed [23]. Goldsmith et al. showed that antipsychotic treatment leads to a decrease in IL-1β, IL-6, IL-4, an increase in sIL-2R, IL-12, and no influence on IFN-γ, TNF-α, IL-2, as well as TGF-β [18]. The Romeo et al. meta-analysis, consisting of 47 studies, 7 of which involved FEP subjects, showed that antipsychotic treatment decreases pro-inflammatory IL-1β and IFN-γ and increases anti-inflammatory sTNF-R2 and sIL-2R. Additionally, a decreasing tendency in IL-6, TNF-α, and IL-4 was reported [22]. The results of aforementioned meta-analyses are not consistent, although all of them indicate a decrease in IL-1β and an increase in sIL-2R, which is coherent with the suggested anti-inflammatory impact of antipsychotic drugs. Studies involving FEP subjects partially confirm these results. A meta-analysis of 7 FEP studies showed that after antipsychotic treatment, IL-1β, IL-6, IL-4 (but not IFN-γ and TNF-α) diminished, which is in contrast to observations from AR studies [22]. Results obtained by Capuzzi et al. are to some extent coherent with prior meta-analyses. After 4 weeks of antipsychotic treatment a decrease in concentration of IL-1β, IL-6 and IL-2 was observed. These 3 interleukines were described as state markers. Similarly, Romeo et al. observed no difference in concentration of TNF-α i IFN-γ in FEP patients [16,18,22]. These markers along with IL-17 were identified as trait markers [22].

In all mentioned meta-analyses, a decrease in IL-1β was reported after antipsychotic treatment. When considering FEP subjects only, both Romeo et al. and Capuzzi et al. observed a significant impact of neuroleptic treatment on decrease in IL-1β and IL-6 levels and no impact on TNF-α and IFN-γ concentrations [22,24].

When compared with meta-analyses of studies involving FEP individuals, and the results we obtained are consistent with Capuzzi et al. and Romeo et al. (decrease in IL-1β and IL-6 levels), with Romeo et al. (decrease in IL-4) and with Capuzzi et al. regarding no impact on IL-17 [22,24]. On the other hand, our results were different on changes of TNF-α and IFN-γ levels. We concluded that after treatment levels of these markers were significantly lower, while both Capuzzi et al., and Romero et al. didn’t observe notable changes in their concentrations. Comparing outcomes of our study with meta-analyses, involving the general population of schizophrenic subjects, our results are consistent when it comes to decrease in IL-1β [16,18,22,23]], IL-6 [16,18,22], IL-4 [18,22], TNF-α and IFN-γ [22], and no difference in IL-2 [18,23]. Our study is the only one which indicates a decrease in anti-inflammatory IL-10 in FEP patients after antipsychotic treatment.

The main difference from previous meta-analyses on the same topic in FEP subjects is the number of studies included (12 vs. 8 and 7 respectively). Inclusion of new studies enabled us to facilitate new comparisons for cytokines’ levels. We decided for our inclusion criteria to be less strict than those applied by Capuzzi et al. [24]. The analyzed studies included mostly, but not only, drug-naïve subjects. As stated in the sensitivity analysis, elimination of previously medicated patients only influenced the significance of TNF-α decrease without effects on the results of IL-1β, IL-6, and IFN-γ. Furthermore, some of the studies involved FEP patients both in the course of schizophrenia as well as in a wide spectrum of psychotic disorders. It is notable that assessing whether a FEP patient is schizophrenic or not may be subject to diagnostic error. We decided to include them, as these disorders can have similar genetic and psychopathological backgrounds [43]. Although providing significant implications for further research, our decisions on liberal criteria resulted in high heterogeneity scores.

Overall, our results indicate a major anti-inflammatory effect of antipsychotic treatment due to decreased concentrations of proinflammatory IL-1β, IL-6, IFN-γ, and TNF-α. We also established a significant decrease in anti-inflammatory cytokines IL-4 and IL-10. It may be a consequence of general amelioration of inflammatory activity. Lower levels of anti-inflammatory markers stemming from antipsychotic treatment may also be a result of its side effects. Tek et al. observed that all antipsychotic drugs, apart from ziprasidone, caused increased body weight in FEP subjects [44]. Moreover, metabolic syndrome and obesity are related to increased pro-inflammatory and decreased anti-inflammatory cytokines’ levels, including IL-4 and IL-10 [45,46]. Russel et al. suggest that patients who have higher levels of inflammation in the early stages of schizophrenia may be at greater risk of developing short-term metabolic abnormalities, dyslipidemia in particular [47]. Our results, as well as previous studies, indicate elevated concentration of pro-inflammatory cytokines IL-1β, IL-6, IFN-γ, and TNF-α in FEP subjects [16,18,19]. Independently of proven anti-inflammatory effects, treatment with several antipsychotic drugs including olanzapine, quetiapine and clozapine is also associated with adverse cardio-metabolic side effects, including weight gain, dyslipidemia, and increased risk of diabetes [48]. These metabolic effects of antipsychotic drugs are associated with increased peripheral inflammation, with elevated levels of IL-6, and macrophage infiltration into adipose tissue [48].

Anti-inflammatory effects of antipsychotic drugs may be explained in two ways. Firstly, as a direct drug action exhibited on the immunological system through a decrease in pro-inflammatory/increase in anti-inflammatory cytokines’ levels or an inhibition of microglial activation [49]. Secondly, this effect can occur indirectly, through amelioration of acute psychological stress. This could lead to decrease in pro-inflammatory factors such as IL-6, IL-1β, and TNF-α [42]. Occurrence of a psychotic episode may act as a major stress factor and contribute to increase in inflammatory markers which subsequently normalize while symptoms ease as a result of administered antipsychotic drugs [22]. Karanikas et al. discovered higher levels of both pro-inflammatory (TNF-α, IL-2, IL-12, IFN-γ) and anti-inflammatory (IL-10) cytokines in the FEP group compared with the Ultra High Risk for psychosis (UHR) matched controls group [50]. It indicates an increased mobilization of both the pro- and anti-inflammatory cytokine networks when positive symptoms intensify, reaching psychosis threshold [51]. In the Borovcanin et al. study, IL-6 levels in FEP patients were found to be significantly higher than in AR subjects. It was explained by duration of illness and possible habituation to psychotic symptoms in individuals that were ill for a longer period. Researchers suggested that psychotic symptoms exacerbation may be a lesser stress factor for AR than FEP patients [30]. These results confirm that cytokines’ abnormalities could be a stress-related marker and positive symptoms a potential stressor that activates inflammation.

Kato et al. suggested that antipsychotics may have neurotrophic, neuroprotective, and therapeutic effects on patients with schizophrenia by reducing microglial inflammation and oxidative stress and cellular reactions that follow [49]. Microglia are known to have various receptors of neurotransmitters, including dopamine D2/3 receptors. Through their inhibition, antipsychotics may directly affect microglia activity [49]. This effect may also be caused indirectly. The blockade of D2 receptors increases the turnover of dopamine in the acute and subacute phase, which is associated with the formation of cytotoxic free radicals and oxidative damage [48]. Dopamine signaling regulates some aspects of microglial function in response to immune stimulation with lipopolysaccharide [52]. The microglial hypothesis assumes an elevation in proinflammatory cytokines, such as IL-1β, IL-6, and TNF-α, which is consistent with our results at baseline in FEP subjects. Kato et al. showed that risperidone significantly inhibits IFN-γ induced microglial activation in vitro [53]. MacDowell et al. confirmed the risperidone effect on microglial function by demonstrating that acute exposure attenuates the inflammatory response induced by systemic administration of lipopolysaccharide in rats [54]. IFN-γ induced microglial activation can also be inhibited by aripiprazole through the suppression of intracellular Ca^2+^ concentrations’ increase in microglia [10]. The Ca^2+^ signaling dysfunction was proposed as a central unifying molecular pathology in schizophrenia [49]. The inhibitory effects of antipsychotics on microglial activation could be caused by modulation of the intracellular cascades, such as mitogen-activated protein kinase (MAPK), protein kinase C (PKC) pathways, and also calcium signaling and nuclear factor kappa-light-chain-enhancer of activated B cells (NF-κB) cascades [49].

Researches showed that the impact of antipsychotic drugs on particular cytokines is substantially related to the therapeutic effect and clinical state in schizophrenia. Romero et al. also confirmed a correlation between differences of IL-6 levels, and positive schizophrenia symptoms’ scores, after antipsychotic treatment [22]. Simsek et al. demonstrated that the concentration of anti-inflammatory interleukin IL-4 and IL-10 was inversely correlated with the intensity of negative symptoms in first schizophrenic episode [51]. The influence of IL-10 levels on increase in negative symptoms was also confirmed by Noto et al. [40]. They demonstrated an inverse correlation between decrease in IL-10, after risperidone treatment, and PANSS negative score, which suggests that IL-10 may be protective against negative symptoms. It was proposed earlier that IL-10 has neuroprotective effects. Arimoto et al. showed that IL-10 protects dopaminergic neurons in the substantia nigra against inflammation-mediated degeneration [55]. Pathophysiology of inflammatory responses may be of crucial importance in the development and treatment of negative and cognitive symptoms. Goldsmith et al.’s findings suggest that the deficit schizophrenia subtype is associated with increased inflammation. TNF-α and IL-6 were associated with the deficit syndrome, and TNF-α predicted blunted affect, alogia, and total negative symptoms in schizophrenia patients [56]. Goldsmith et al. also reported that TNF-α and IL-6 may predict development of negative symptoms (TNF-α irrespective of baseline depressive symptoms) in individuals at clinically high-risk of psychosis [57]. Moreover, chronic peripheral inflammation may be associated with cognitive impairment in schizophrenia. The Multicentric FACE-SZ Dataset demonstrated that abnormal CRP levels in schizophrenic individuals were associated with a decline in cognitive functions, such as working and semantic memory, learning abilities, mental flexibility, visual attention, and speed of processing [58].

## 5. Limitations

Our work has several limitations. First of all, we used liberal inclusion criteria. We did not discriminate studies with patients already treated for short periods during first cytokine measurement and studies on FEP patients with diagnoses other than schizophrenia. The included studies provided measurements at different time-points (4 weeks–7 months). This is a methodological limitation, but also it resulted in high heterogeneity of our sample when compared to other recent publications. Secondly, the results of any meta-analysis (including ours) conducted on mean results extracted from separate studies, should be approached with caution, we did not have access to individual patient data. Finally, we excluded studies not published in English. This might come as an advantage when considering data quality, but it also significantly decreased potential statistical power of our results.

Moreover, our meta-analysis included studies that assessed biomarkers only peripherally and not in cerebrospinal fluid (CSF), which could have a more direct link with the CNS inflammation processes. However, some studies suggest that blood-based biomarkers are valid measures for investigating psychosis. Coughlin et al. demonstrated that CSF levels of IL-6 were significantly correlated with the levels of IL-6 in serum of patients with recent onset of schizophrenia [56]. In spite of such results, we cannot rule out the influence of certain clinical factors, such as obesity, on variations between peripheral and central concentrations of cytokines.

Another limitation of this meta-analysis is the lack of sufficient data regarding potential confounding factors that may affect inflammation processes in the majority of analyzed studies. Obesity, other medical comorbidities, psychoactive substances, as well as tobacco use can influence inflammation markers independently from psychosis and its treatment [16]. Only six of the analyzed studies took into account patients’ body mass index (BMI), only three excluded overweight or obese patients, and only one considered the influence of changing body mass on cytokines levels in the course of antipsychotic treatment. Only seven studies included smoking as a factor possibly influencing inflammation markers. A study of chronic schizophrenia patients, on long-term treatment with antipsychotics, revealed that proinflammatory IL-2 and IL-6 were significantly lower in smokers than nonsmokers, which may be associated with nicotine-induced suppression of some inflammatory cytokines [59]. In 7 out of 12 studies, psychoactive substance use was considered, or only diagnosed addicts were excluded, without assessing the impact of recreational use. Authors of three studies did not evaluate the influence of potential medical comorbidities, some authors excluded only subjects with history of autoimmune diseases.

Establishing how different antipsychotics might influence cytokine levels in FEP subjects, with consideration of factors which could impact inflammation markers, is crucial for obtaining fully reliable conclusions. As Miller and Goldsmith (2017) noticed, further studies should be rigorously designed to consider factors known to influence immune function such as age, sex, race, smoking, BMI, socioeconomic status, insomnia, fasting status and dietary factors, level and type of psychopathology, as well as genetic heterogeneity [60].

The variety of used antipsychotic drugs in analyzed studies is another factor contributing to heterogeneity. As was reported by Romeo et al. (2018), various drugs can have different impacts on levels of inflammation biomarkers [22]. The broadest influence was observed for risperidone, which corresponded with a significant decrease in IL-1β, IL-2, IL-4, IL-6, IL-10, and increase in IL-12 levels. Treatment with olanzapine resulted in IL-2 and IFN-γ decrease, haloperidol—IL-2 decrease, aripiprazole—IL-10 increase. Quetiapine administration didn’t influence cytokines levels, while treatment with clozapine resulted in an anti-inflammatory effect related to increase in sTNF-R1, sTNF-R2 and sIL-2R, as well as pro-inflammatory—increase in IL-6 [22]. Different types of antipsychotic drugs have varying influence on the serotoninergic and noradrenergic systems. This could have an impact on the hypothalamic–pituitary–adrenal axis and, subsequently, inflammation biomarkers in stress response [22]. Unfortunately, the available data was not sufficient to examine the impact of particular drugs on biomarkers in FEP subjects. Not all of the authors precisely stated what drugs were in the scope of “antipsychotic treatment.” In 6 out of 12 studies, patients were treated with one antipsychotic drug—risperidone, however, it was not always in monotherapy (benzodiazepines, anticholinergics, and antidepressants were additionally used in some studies). In the remaining studies, various drugs were administered, and no subgroup analysis was performed. Assessing how different antipsychotics influence cytokines’ levels in FEP subjects, and how it correlates with clinical response, would be important for further studies.

Despite these limitations and resulting high heterogeneity, our meta-analysis provides further evidence for the importance of antipsychotic treatment and its role in inflammation mechanisms in FEP subjects. Neuroleptics seem to exert an impact on a wide range of cytokines with significant anti-inflammatory effect. It is relevant for the hypothesis that a subset of subjects may manifest a schizophrenia immunophenotype [16,18]. Immune treatments may, therefore, have potential in schizophrenia therapy. Studies of the efficacy and tolerability of nonsteroidal anti-inflammatory drugs (NSAIDs) in schizophrenia indicate significant clinical improvement [61]. Moreover, several small trials support the feasibility and potential efficacy of adjunctive monoclonal antibody immunotherapy. The addition of recombinant human IFN-γ, as well as anti-IL-6 receptor monoclonal antibody (tocilizumab), was well tolerated and led to clinical improvement [62,63]. Monoclonal antibody immunotherapy seems to be a more preferable option in schizophrenia treatment when compared to anti-inflammatory agents, since monoclonal antibodies act on specific inflammatory cytokines, as opposed to anti-inflammatory agents, which do have significant off-target effects [60]. Larger trials of adjunctive monoclonal antibody immunotherapy would be an important contribution to novel methods of schizophrenia treatment. Measurement of cytokines’ levels before and after treatment can aid in determining schizophrenia patients’ subpopulations in which the potential adjunct of novel immunomodulatory agents could lead to improvement in therapeutic effects. Miller and Goldsmith (2017) reported that measurement of immune activation overall patterns, such as flow cytometry of macrophages/monocytes vs. Th1 vs. Th2 vs. Th17 cells, may be more important than the assessment of individual cytokines or other markers [60]. In order to improve our understanding of the biological mechanisms of the immune response after antipsychotic treatment, it is important to study both the influence of a wide range of confounding factors and patient-specific symptoms, as well as cognitive features.

## Figures and Tables

**Figure 1 jcm-10-02488-f001:**
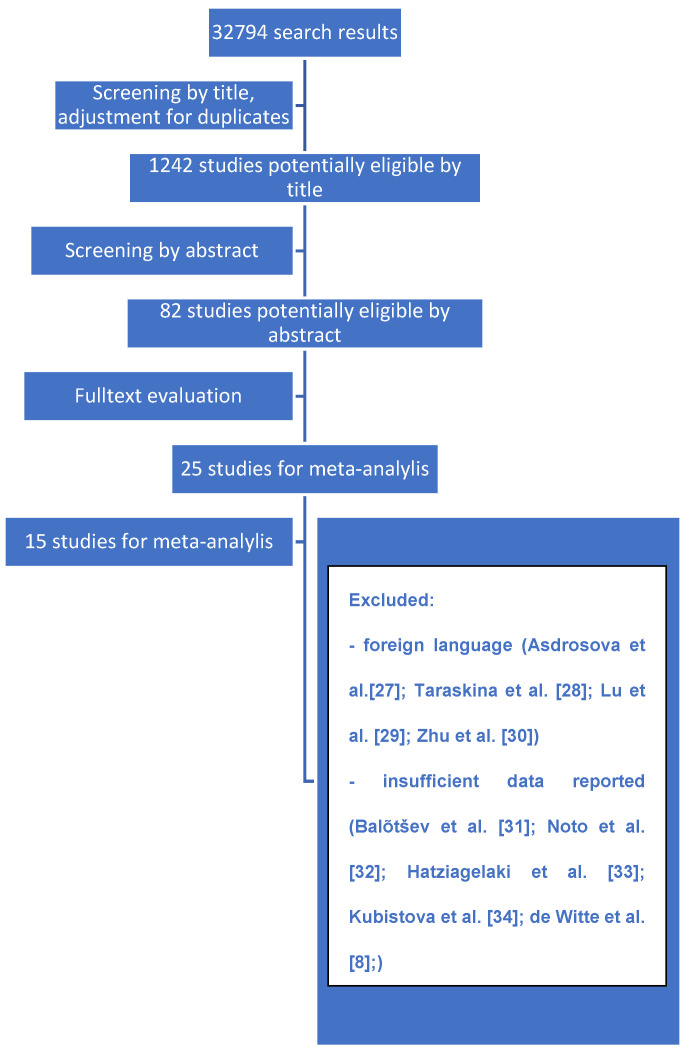
Study selection—flow diagram.

**Figure 2 jcm-10-02488-f002:**
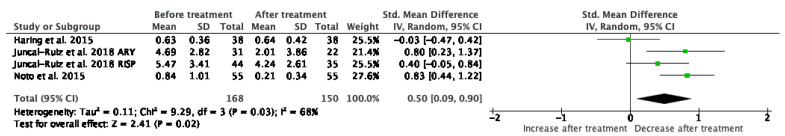
Meta-analysis of IL-10.

**Figure 3 jcm-10-02488-f003:**
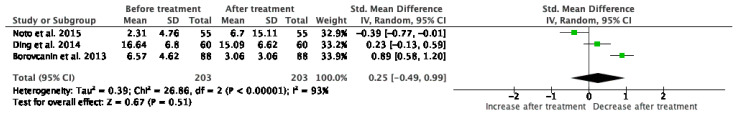
Meta-analysis of IL-17.

**Figure 4 jcm-10-02488-f004:**
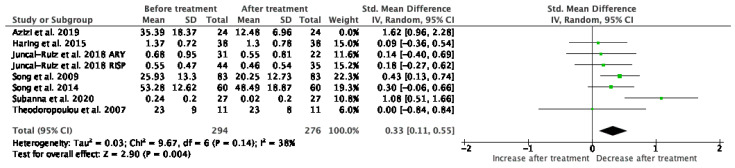
Meta-analysis of IL-1β.

**Figure 5 jcm-10-02488-f005:**
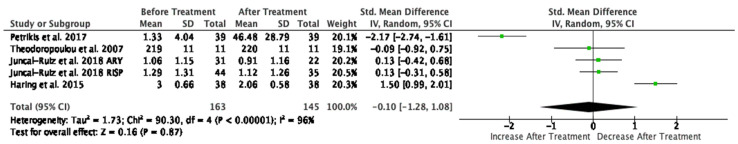
Meta-analysis of IL-2.

**Figure 6 jcm-10-02488-f006:**
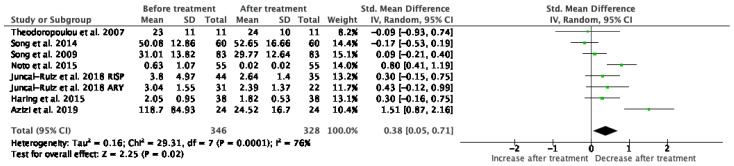
Meta-analysis of TNF-α.

**Figure 7 jcm-10-02488-f007:**
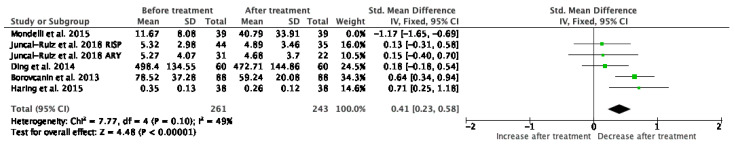
Meta-analysis of IFN-γ.

**Figure 8 jcm-10-02488-f008:**
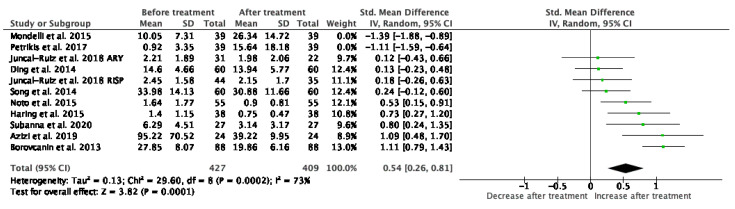
Meta-analysis of IL-6.

**Figure 9 jcm-10-02488-f009:**
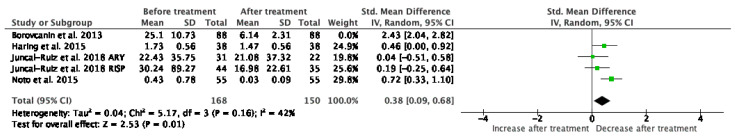
Meta-analysis of IL-4.

**Table 1 jcm-10-02488-t001:** Meta-analysis characteristics.

Cytokine	Number of Studies	N before/after Treatment	SMD	95% CI	*p*	I^2^	Studies
IL-1β	7	294/276	0.43	0.13, 0.74	0.006	74%	Juncal-Ruiz 2018 [32], Haring 2015 [33], Song 2014 [34] Subanna 2020 [35], Theodoropoulou 2007 [36], Song 2009 [37], Azizi 2019 [38]
IL-2	4	163/145	−0.1	−1.12, 0.93	0.85	96%	Petrikis 2017 [39], Juncal-Ruiz 2018 [32], Haring 2015 [33], Theodoropoulou 2001 [36]
IL-4	4	168/150	0.38	0.09, 0.68	0.01	42%	Noto 2015 [40], Juncal-Ruiz 2018 [32], Haring 2015 [33], Borovcanin 2013 [30]
IL-6	10	427/409	0.54	0.26, 0.81	0.0001	73%	Noto 2015 [40], Petrikis 2017 [39], Juncal-Ruiz 2018 [32], Haring 2015 [33], Song 2014 [34], Subanna 2020 [35], Ding 2014 [41], Mondelli 2015 [42], Borovcanin 2013 [30], Azizi 2019 [38]
IL-10	3	168/150	0.5	0.09, 0.90	0.02	68%	Noto 2015 [40], Juncal-Ruiz 2018 [32], Haring 2015 [33]
IL-17	3	203	0.29	−0.49, 0.99	0.51	93%	Noto 2015 [40], Ding 2014 [41], Borovcanin 2013 [30]
IFN-γ	5	261/243	0.38	0.13, 0.64	0.003	49%	Juncal-Ruiz 2018 [32], Haring 2015 [33], Ding 2014 [41], Mondelli 2015 [42], Borovcanin 2013 [30]
TNF-α	7	346/328	0.35	0.03, 0.67	0.03	78%	Juncal-Ruiz 2018 [32], Haring 2015 [33], Song 2014 [34], Noto 2015 [40], Theodoropoulou 2001 [36], Song 2009 [37], Azizi 2019 [38]

**Table 2 jcm-10-02488-t002:** Characteristics of included studies.

Study	Duration	Cytokines	Medication	Sample	*N*
Noto, 2015 [40]	10 weeks	IL-1b, Il-6, IL-4, IL-10	Risperidone	serum	55
Petrikis, 2017 [39]	6 weeks	IL-2, IL-6, TGF-β2	Olanzapine, Risperidone, Aripiprazole, Quetiapine, Haloperidol	serum	39
Juncal-Ruiz, 2018 [32]	12 weeks	IL-6 IL-1β TNF-α IFN-γ IL10 IL-17α IL-13 IL-12 IL-2 IL-21 IL-4 IL-23 IL-5 IL-7 IL-8 MIP-1α MIP-1β MIP-3α ITAC GMCSF Fractalkine	Aripiprazole, Risperidone	serum	75
Haring, 2015 [33]	28 weeks	IL-2, IL-4, IL-6, IL-8, IL-10, VEGF, IFN-γ, TNF-α, IL-1α, IL-1β, MCP-1, EGF,	No data	serum	38
Song, 2014 [34]	2, 4, 8, 12, 24 weeks	IL-1β, TNF-α, IL-6	Risperidone	serum	62
Azizi, 2019 [38]	12 weeks	TNF-α, IL-1β, IL6	Risperidone	serum	24
Song, 2009 [37]	4 weeks	TNF-α, IL-1β	Risperidone	serum	83
Subbanna, 2020 [35]	12 weeks	IL-1β, IL-6, IL-17A, IL-23, IL-33	No data	plasma	27
Ding 2014 [41]	4 weeks	IL-17, IFN-γ, IL-6	Risperidone	plasma	69
Mondelli 2015 [42]	12 weeks	IFN-γ, IL-6	Olanzapine, Risperidone, Quetiapine, Aripiprazole.	serum	39
Theodoropoulou 2001 [36]	4 weeks	TNF-α, IL-2, IL-1β	No data	serum	53
Borovcanin 2013 [30]	4 weeks	IFN-γ, IL-17, IL-4, IL-6, IL-27, TGF-β	No data	serum	78

**Table 3 jcm-10-02488-t003:** Risk of bias: 1—Low risk of bias; 2—Moderate risk of bias; 3—Serious risk of bias; 4—Critical risk of bias; 0—No information.

Study	Bias Due to Confounding	Bias in Selection of Participants into the Study	Bias in Classification of Interventions	Bias Due to Deviations from Intended Interventions	Bias Due to Missing Data	Bias in Measurement of the Outcome	Bias in Selection of the Reported Result
Noto, 2015 [40]	2	1	1	0	1	1	2
Petrikis, 2017 [39]	2	3	1	0	1	1	1
Juncal-Ruiz, 2018 [32]	2	1	1	0	1	1	1
Haring, 2015 [33]	2	2	1	0	1	1	1
Song, 2014 [34]	1	1	1	0	1	1	1
Azizi, 2019 [38]	2	2	1	0	1	1	2
Song, 2009 [37]	1	1	1	0	1	1	1
Subbanna, 2020 [35]	1	2	1	0	1	1	2
Ding 2014 [41]	1	1	1	2	1	1	1
Mondelli 2015 [42]	2	1	1	0	1	1	1
Theodoropoulou 2001 [36]	3	1	1	1	1	1	1
Borovcanin 2013 [30]	3	2	1	1	1	1	1

## Data Availability

Not applicable.

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
