# Peer review of "A Meta-Analysis of the Influence of Antipsychotics on Cytokines Levels in First Episode Psychosis"

_jcm, 2021, doi:10.3390/jcm10112488_

Round 1

Reviewer 1 Report

Marcinowicz et al. found blood levels of IL-b, IL-6, IFN-g, TNF-a, IL-4 and IL-10 were decreased after antipyschotics treatment in patients with first episode psychosis using a meta-analysis. Although it is an interesting manuscript, several issues should be addressed.

  1. The authors should follow PRISMA to report a meta-analysis.
  2. The authors should provide the definition of first episode psychosis.
  3. Were there any differences between serum and plasma?
  4. In line 141, what was the Borovcanin et al. results?
  5. There were two Table 1s.
  6. The authors should provide clear Figures.
  7. The authors should perform the sensitivity analysis.
  8. The authors should provide the results of publication bias.
  9. Limited studies were included in the meta-analysis. There were significant heterogeneities among studies. The authors should discuss these limitations.
  10. English editing should be performed on this manuscript.

Author Response

Dear Reviewers,

We made all requested changes to the best of our ability.

We hope the current version meets Your expectations. Nevertheless we wait for further instructions if any more revisions are needed.

Best regards

Reviewer 2 Report

Marcinowicz et al performed a meta-analysis to investigate the effects of antipsychotics on cytokine levels in FEP patients. Statistical analyses were appropriate and study limitations were thoroughly discussed. However, in order for the manuscript to publish, the organization and writing should be improved. Specific comments are below:

  • There are two tables labeled as “Table 1”.
  • I suggest that the authors present Table showing all studies analyzed first, and include a paragraph describing the table as the first section under Results.
  • All the tables and Figures should have captions. For example, what the columns in Table 1 on page 2? Abbreviations should be spelled out in the caption. And there are two columns of “P”; what are they?  
  • There are many careless typos and grammatical errors. For example: Line 151: “which effects vary from others”; Line 152: “several analyzes were performed”; Line 170: “study analize.”

Author Response

(The authors gave the same response as above.)

Round 2

Reviewer 1 Report

The authors should show corrections in both the text and response letter.

  1. The authors should provide PRISMA Flow Diagram.
  2. The authors should indicate whether serum or plasma was used in each study in Table 3. They also should assess whether there were differences between serum and plasma.
  3. Limited studies were included in the meta-analysis. There were significant heterogeneities among studies. Although the authors claimed these limitations were thoroughly discussed, this reviewer was not able to find that.

Author Response

1. The authors should provide PRISMA Flow Diagram.
 attached
2. The authors should indicate whether serum or plasma was used in each study in Table 3.
They also should assess whether there were differences between serum and plasma.
 Only two studies used plasma (Subbanna and Ding, the rest used serum). Sensitivity
analysis did not suggest any relevance of this difference. Table 3 has been updated.
3. Limited studies were included in the meta-analysis. There were significant heterogeneities
among studies. Although the authors claimed these limitations were thoroughly discussed, this
reviewer was not able to find that.
 We excluded several studies mostly because they were not published in
English or did not report data in a feasible form. Should we describe the
process of each study exclusion and cite these studies in bibliography?
 We reported possible causes of heterogeneity – not all patients were strictly
drug-naïve (441), the treatment period varied between studies (443) so we
had to make an assumption that the relation between treatment time and
cytokine levels is linear, some studies included FEP patients and others “first
episode schizophrenics, which suggests that the psychotic features might
have been more schizophrenia specific. In lines 459-473 we consider the
heterogeneity with regard to potential confounding factors such as obesity,
substance abuse, smoking and not excluded autoimmune diseases etc. In
lines 480-499 we discuss that there was much impurity regarding
antipsychotic drugs used by researchers, only 6 studies utilized a one drug
design.
 Should we discuss it in more thorough detail or is it lacking some other
important issues with regard to heterogeneity?

Best regards
Piotr Marcinowicz

Round 3

Reviewer 1 Report

The authors have carefully taken all comments.